# STYLE-BASED ENCODER PRE-TRAINING FOR MULTI-MODAL IMAGE SYNTHESIS

## ABSTRACT

Image-to-image (I2I) translation aims to translate images from one domain to another. To tackle the multi-modal version of I2I translation, where input and output domains have a one-to-many relation, an extra latent input is provided to the generator to specify a particular output. Recent works propose involved training objectives to learn a latent embedding, jointly with the generator, that models the distribution of possible outputs. Alternatively, we study a simple, yet powerful pre-training strategy for multi-modal I2I translation. We first pre-train an encoder, using a proxy task, to encode the style of an image, such as color and texture, into a low-dimensional latent style vector. Then we train a generator to transform an input image along with a style-code to the output domain. Our generator achieves state-of-the-art results on several benchmarks with a training objective that includes just a GAN loss and a reconstruction loss, which simplifies and speeds up the training significantly compared to competing approaches. We further study the contribution of different loss terms to learning the task of multi-modal I2I translation, and finally we show that the learned style embedding is not dependent on the target domain and generalizes well to other domains.

## 1 INTRODUCTION

Image-to-Image (I2I) translation is the task of transforming images from one domain to another (e.g., semantic maps → scenes, sketches → photo-realistic images, etc.). Many problems in computer vision and graphics can be cast as I2I translation, such as photo-realistic image synthesis (Chen & Koltun (2017); Isola et al. (2017); Wang et al. (2018a)), super-resolution (Ledig et al. (2017)), colorization (Zhang et al. (2016; 2017a)), and inpainting (Pathak et al. (2016)). Therefore, I2I translation has recently received significant attention in the literature. One main challenge in I2I translation is the multi-modal nature for many such tasks – the relation between an input domain $A$ and an output domain $B$ is often times one-to-many, where a single input image $I_i^A \in A$ can be mapped to different output images from domain $B$. For example, a sketch of a shoe or a handbag can be mapped to corresponding objects with different colors or styles, or a semantic map of a scene can be mapped to many scenes with different appearance, lighting and/or weather conditions. Since I2I translation networks typically learn one-to-one mappings due to their deterministic nature, an extra input is required to specify an output mode to which an input image will be translated. Simply injecting extra random noise as input proved to be ineffective as shown in (Isola et al. (2017); Zhu et al. (2017b)), where the generator network just learns to ignore the extra noise and collapses to a single or few modes (which is one form of the mode collapse problem). To overcome this problem, Zhu et al. (2017b) proposed *BicycleGAN*, which learns to encode the distribution of different possible outputs into a latent vector $z$, and then learns a deterministic mapping $G : (A, z) \rightarrow B$. So, depending on the latent vector $z$, a single input $I_i^A \in A$ can be mapped to multiple outputs in $B$. While BicycleGAN requires paired training data, several works (Lee et al. (2018); Huang et al. (2018)) extended it to the unsupervised case, where images in domains $A$ and $B$ are not in correspondence ('unpaired'). One main component of unpaired I2I is a cross-cycle consistency constraint, where the network generates an intermediate output by swapping the styles of a pair of images, then swaps the style between the intermediate output again to reconstruct the original images. This enforces that the latent vector $z$ preserves the encoded style information when translated from an image $i$ to another image $j$ and back to image $i$ again. This constraint can also be applied to paired training data, where it encourages style/attribute transfer between images. However, training BicycleGAN (Zhu et al. (2017b)) or its unsupervised counterparts (Huang et al. (2018); Lee et al. (2018)) is not trivial. For example,

BicycleGAN combines the objectives of both conditional Variational Auto-Encoders (cVAEs) (Sohn et al. (2015)) and a conditional version of Latent Regressor GANs (cLR-GANs) (Donahue et al. (2016); Dumoulin et al. (2016)) to train their network. The training objective of (Huang et al. (2018); Lee et al. (2018)) is even more involved to handle the unsupervised setup.

In this work, we aim to simplify the training of general purpose multi-modal I2I translation networks, while also improving the diversity and expressiveness of different styles in the output domain. Our approach is inspired by the work of Meshry et al. (2019) which utilizes a staged training strategy to re-render scenes under different lighting, time of day, and weather conditions. We propose a pretraining approach for style encoders, in multi-modal I2I translation networks, which makes the training simpler and faster by requiring fewer losses/constraints. Our approach is also inspired by the standard training paradigm in visual recognition of first pretraining on a proxy task, either large supervised datasets (e.g., ImageNet) (Krizhevsky et al. (2012); Sun et al. (2017); Mahajan et al. (2018)) or unsupervised tasks (e.g., Doersch et al. (2015); Noroozi & Favaro (2016)), and then fine-tuning (transfer learning) on the desired task. Similarly, we propose to pretrain the encoder using a proxy task that encourages capturing style into a latent space. Our goal is to highlight the importance of pretraining for I2I networks and demonstrate that a simple approach can be very effective for multi-modal image synthesis. In particular, we make the following contributions:

- We explore style pretraining and its generalization for the task of multi-modal I2I translation, which simplifies and speeds up the training compared to competing approaches.

- We provide a study of the importance of different losses and regularization terms for multi-modal I2I translation networks.

- We show that the pretrained latent embeddings is not dependent on the target domain and generalizes well to other domains (transfer learning).

- We achieve state-of-the art results on several benchmarks in terms of style capture and transfer, and diversity of results.

## 2 RELATED WORK

**Deep generative models** There has been incredible progress in the field of image synthesis using deep neural networks. In its unconditional setting, a decoder network learns to map random values drawn from a prior distribution (typically Gaussian) to output images. Variational Auto-Encoders (VAEs) (Kingma & Welling (2014)) assume a bijection mapping between output images and some latent distribution and learn to map the latent distribution to a unit Gaussian using the reparameterization trick. Alternatively, Generative Adversarial Networks (GANs) (Goodfellow et al. (2014)) directly map random values sampled from a unit Gaussian to images, while using a discriminator network to enforce that the distribution of generated images resembles that of real images. Recent works proposed improvements to stabilize the training (Gulrajani et al. (2017); Karnewar & Iyengar (2019); Mao et al. (2017); Radford et al. (2016)) and improve the quality and diversity of the output (Karras et al. (2018; 2019)). Other works combine both VAEs and GANs into a hybrid VAE-GAN model (Larsen et al. (2016); Rosca et al. (2017)).

**Conditional image synthesis** Instead of generating images from input noise, the generator can be augmented with side information in the form of extra conditional inputs. For example, Sohn et al. (2015) extended VAEs to their conditional setup (cVAEs). Also, GANs can be conditioned on different information, like class labels (Mirza & Osindero (2014); Odena et al. (2017); Van den Oord et al. (2016)), language description (Mansimov et al. (2016); Reed et al. (2016)), or an image from another domain (Chen & Koltun (2017); Isola et al. (2017)). The latter is called Image-to-Image translation.

**Image-to-Image (I2I) translation** I2I translation is the task of transforming an image from one domain, such as a sketch, into a another domain, such as photo-realistic images. While there are regression-based approaches to this problem (Chen & Koltun (2017); Hoshen & Wolf (2018)), significant successes in this field are based on GANs and the influential work of pix2pix (Isola et al. (2017)). Following the success of pix2pix (Isola et al. (2017)), I2I translation has since been utilized in a large number of tasks, like inpainting (Pathak et al. (2016)), colorization (Zhang et al. (2016; 2017a)), super-resolution (Ledig et al. (2017)), rendering (Martin-Brualla et al. (2018); Meshry et al. (2019); Thies et al. (2019)), and many more (Dong et al. (2017); Wang & Gupta (2016); Zhang et al. (2017b)). There has also been works to extend this task to the unsupervised setting (Hoshen & Wolf

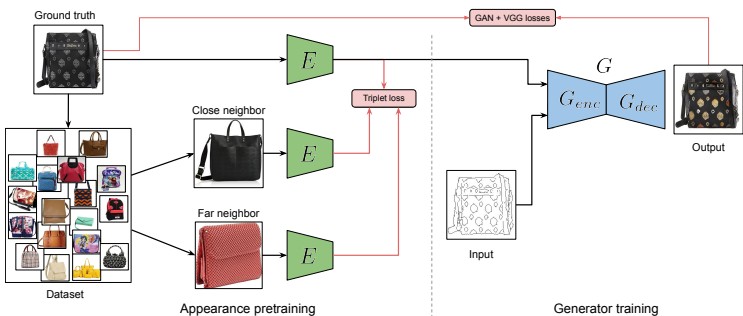

Figure 1: Overview of our training pipeline. Stage 1: pretraining the style encoder $E$ using a triplet loss. Stages 2, 3: training the generator $G$, and finetuning both $G, E$ together using *GAN* and reconstruction losses.

(2018); Kim et al. (2017); Liu et al. (2017); Ma et al. (2019); Royer et al. (2017); Zhu et al. (2017a)), to multiple domains (Choi et al. (2018)), and to videos (Chan et al. (2018); Wang et al. (2018b)).

**Multi-modal I2I translation** Image translation networks are typically deterministic function approximators that learn a one-to-one mapping between inputs and outputs. To extend I2I translation to the case of diverse multi-modal outputs, Zhu et al. (2017b) proposed the BicycleGAN framework that learns a latent distribution that encodes the variability in the output domain and conditions the generator on this extra latent vector for multi-modal image synthesis. Wang et al. (2018a;b) learn instance-wise latent features for different objects in a target image, which are clustered after training to find $f$ fixed modes for different semantic classes. At test time, they sample one of the feature clusters for each object to achieve multi-modal synthesis. Other works extended the multi-modal I2I framework to the unpaired settings where images from the input and output domains are not in correspondence (Almahairi et al. (2018); Huang et al. (2018); Lee et al. (2018)) by augmenting BicycleGAN with different forms of a cross-cycle consistency constraint between two unpaired images. In our work, we focus on the supervised setting of multi-modal I2I translation. We propose a simple, yet effective, pretraining strategy to learn a latent distribution that encodes variability in the output domain. The learned distribution can be easily adapted to new unseen datasets with simple fine-tuning, instead of training from random initialization.

## 3 APPROACH

Current multi-modal image translation networks require an extra input $z$ that allows for modelling the one-to-many relation between an input domain $A$ and an output domain $B$ as a one-to-one relation from a pair of inputs $(A, z) \rightarrow B$. In previous approaches, there has been a trade-off between simplicity and effectiveness for providing the input $z$. On one hand, providing random noise as the extra input $z$ maintains a simple training objective (same as in pix2pix (Isola et al. (2017))). However, Isola et al. (2017); Zhu et al. (2017b) showed that the generator has little incentive to utilize the input vector $z$ since it only encodes random information, and therefore the generator ends up ignoring $z$ and collapsing to one or few modes. On the other hand, *BicycleGAN* (Zhu et al. (2017b)) combines the objectives of both conditional Variational Auto-Encoder GANs (cVAE-GAN) and conditional Latent Regressor GANs (cLR-GAN) to learn a latent embedding $z$ simultaneously with the generator $G$. Their training enforces two cycle consistencies: $B \rightarrow z \rightarrow \hat{B}$ and $z \rightarrow \tilde{B} \rightarrow \hat{z}$. This proved to be very effective, but the training objective is more involved, which makes the training slower. Also, since the latent embedding is being trained simultaneously with the the generator, hyper-parameter tuning becomes more critical and sensitive.

We aim to combine the best of both worlds: an effective training of a latent embedding that models the distribution of possible outputs, while retaining a simple training objective. This would allow for faster and more efficient training, as well as less sensitivity to hyper-parameters. We observe that the variability in many target domains can be represented by the style diversity of images in the target domain $B$, where the style is defined in terms of the gram matrices used in the Neural Style Transfer literature (Gatys et al. (2016)). Then, we learn an embedding by separately training an encoder network $E$ on an auxiliary task to optimize for $z = E(I^B)$ capturing the style of an image $I^B$. Finally, since we now have learned a deterministic mapping between $z$ and the style of the target

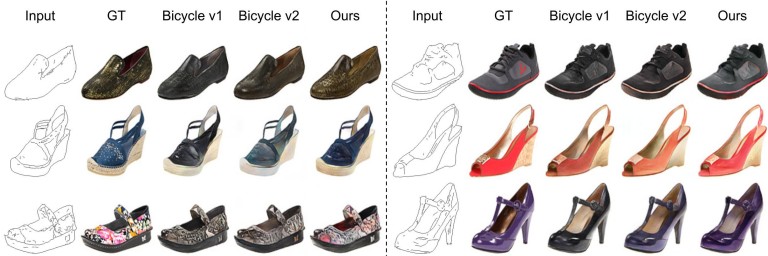

Figure 2: Qualitative comparison with baselines. Ours better matches the ground truth (GT) style.

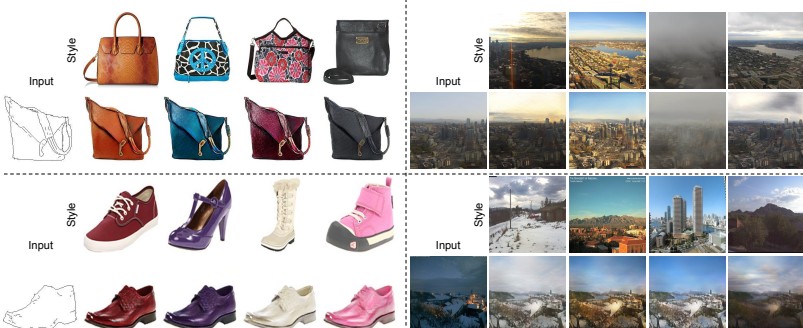

Figure 3: Style transfer for different datasets. For each dataset, we show output for applying different styles to the same input image.

output image $I^B$, training the generator $G$ becomes simpler as $G$ is just required to discover the correlation between output images and their corresponding style embedding $z$.

To incorporate this into BicycleGAN (Zhu et al. (2017b)), we replace the simultaneous training of the encoder $E$ and the generator $G$ with a staged training as follows:

**Stage 1:** Pretrain $E$ on a proxy task that optimizes an embedding of images in the output domain $B$ into a low-dimensional style latent space, such that images with similar styles lie closely in that space (i.e. clustered).

**Stage 2:** Train the generator network $G$ while fixing the encoder $E$, so that $G$ learns to associate the style of output images to their deterministic style embedding $z = E(I_B)$.

**Stage 3:** Fine-tune both the $E$ and $G$ networks together, allowing for the style embedding to be further adapted to best suit the image synthesis task for the target domain.

The intuition why such staged training would be effective for multi-modal I2I translation is that the encoder is pretrained to model different modes of the output distribution as clusters of images with similar styles (refer to the supp. material, figures 6,7, for a visualization of pretrained latent embeddings). During stage 2, the latent space is kept fixed, and the input latent to the generator can be used to clearly distinguish the style cluster to which the output belongs, which makes the multi-modal synthesis task easier for the generator. Finally, stage 3 finetunes the learned embedding to better serve the synthesis task at hand. Next, we explain how to pre-train the style encoder network $E$ in Section 3.1, and how to train the generator $G$ using the pre-learned embeddings (Section 3.2). Finally, we demonstrate the generalization of pre-training the style encoder $E$ in Section 3.3.

## 3.1 PRE-TRAINING THE STYLE ENCODER $E$

The goal of pre-training the encoder network $E$ is to learn a deterministic mapping from the style of a target image $I_i^B \in B$ to a latent style code $z_i = E(I_i^B)$. Ideally, images with similar styles should be close in the style embedding space, while images with different styles should be far from each other. To supervise training such embedding, we utilize the style loss (Gatys et al. (2016)) as a distance metric to measure the style similarity between any two given images. The style encoder network $E$ is then trained using a triplet loss (Schroff et al. (2015)), where the input is a triplet of images $(I_a, I_p, I_n)$, where $(I_a, I_p)$ have similar style, while $(I_a, I_n)$ have different style, as measured by the style loss metric. The training objective for $E$ is given by:

$$\mathcal{L}^{\text{tri}}(I_a, I_{\text{p}}, I_{\text{n}}) = \max\left(\left[\|z_a - z_p\|^2 - \|z_a - z_n\|^2 + \alpha\right], 0\right) + \lambda \mathcal{L}^{\text{reg}}(z_a, z_p, z_n) \tag{1}$$

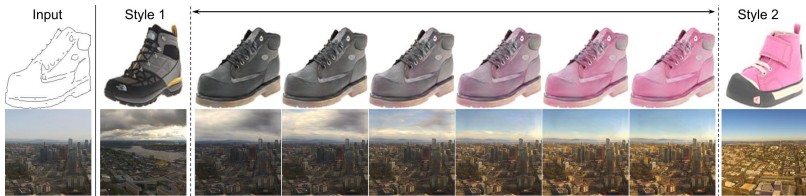

Figure 4: Style interpolation. Left column is the input to the generator $G$, second and last columns are input style images to the style encoder, and the middle images are linear interpolation in the embedding space.

where $\alpha$ is a separation margin, $\lambda$ is a relative weighting parameter between the main triplet objective and an optional regularization term $\mathcal{L}^{\text{reg}}(\cdot)$ which is an $L2$ regularization to encourage learning a compact latent space.

**Triplet selection.** To generate triplets for pre-training the encoder $E$, we compute the set of $k_c$ closest and $k_f$ furthest neighbors for each anchor image $I_a$ as measured by the style loss. Then, for each anchor image $I_a$, we randomly sample a positive image $I_p$ and a negative image $I_n$ from the set of closest and furthest neighbors respectively. We found that, for large datasets, it is sufficient to generate triplets for a subset of the training images. One challenge is the set of images with an outlier style. Such images will be furthest neighbors to most images, and can mislead the training by just projecting outlier images to separate clusters. To deal with this, we sample the negative style image $I_n$ from a larger set of furthest neighbors; while the positive image $I_p$ is sampled from a small set of closest neighbors so that it would have reasonable style similarity to the anchor image.

## 3.2 GENERATOR TRAINING

After pre-training the style encoder $E$ (stage 1), we have established a mapping from images in the output domain, $I^B \in B$, to their style-embedding $z = E(I^B)$. Feeding the style embedding as input to the generator during training, the generator has good incentive to associate the style of output images to their corresponding style embedding instead of learning to hallucinate the style. It's important to retain the deterministic correspondence between images and their style codes to facilitate the job of the generator to discover this correlation. This is why, during stage 2, we keep the weights of the style encoder $E$ fixed. The forward pass reconstructs a training image $I_i^B$ as $\hat{I}_i^B = G(I_i^A, z_i)$, where $z_i = E(I_i^B)$. The training objective is similar to that of pix2pix (Isola et al. (2017)),

$$\mathcal{L}^{\text{img}}(I_i^B, \hat{I}_i^B) = \mathcal{L}_{\text{cGAN}}(I_i^B, \hat{I}_i^B) + \lambda_{\text{rec}} \mathcal{L}_{\text{rec}}(I_i^B, \hat{I}_i^B) \tag{2}$$

where we use the Least Square GAN loss (LSGAN) (Mao et al. (2017)) for the $\mathcal{L}_{\text{cGAN}}$ term, and a VGG-based perceptual loss (Johnson et al. (2016)) for the reconstruction term $\mathcal{L}_{\text{rec}}$. Once the generator has learned to associate the output style with the input style embedding, stage 3 fine-tunes both the generator $G$ and the style encoder $E$ together using the same objective (2).

**Style sampling.** To perform multimodal synthesis on a given input at test time, we can capture the latent vector $z$ from any existing image and transfer the style to the generated image. However, if we wish to sample styles directly from the latent distribution, we can optionally enforce a prior on the latent distribution. For example, we can add a KL divergence term on the latent vectors to enforce a unit Gaussian prior. In our experiments, we found it more effective to add an $L2$ regularization on the the latent vectors to enforce zero-mean embeddings and limit the variance of the latent space. We then compute an empirical standard deviation for sampling. Another alternative to enable sampling is to train a mapper network $\mathcal{M}$ to map the unit Gaussian to the latent distribution. This can be done as a post-processing step after the style encoder has been trained and finetuned. Specifically, we follow the nearest-neighbor based Implicit Maximum Likelihood Estimation (IMLE) training (Li & Malik (2018); Hoshen et al. (2019)) to train the mapper network $\mathcal{M}$. The training objective is given by:

$$\mathcal{M} = \arg\min_{\tilde{\mathcal{M}}} \sum_i \|z_i - \tilde{\mathcal{M}}(e_i)\|_2^2, \quad e_i = \arg\min_{r_j} \|z_i - \mathcal{M}(r_j)\|_2^2 \tag{3}$$

where $\{r_j\}$ is a set of random samples from the unit Gaussian prior, and for each latent code $z_i$, we select $e_i$ that maps to the nearest neighbor $\mathcal{M}(e_i)$ to $z_i$.

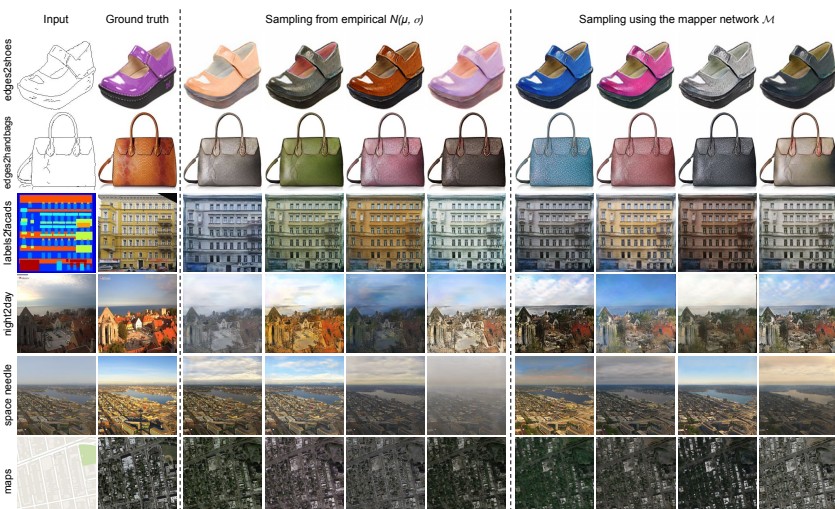

Figure 5: Style sampling for different datasets using our approach v3. We sample either from $N(\mu, \sigma)$, where $\mu, \sigma$ are computed from the train set (middle), or using the mapper network $\mathcal{M}$ (right).

### 3.3 GENERALIZING THE PRE-TRAINING STAGE

The goal of the style pretraining is to learn an embedding that mimics the style loss, where images with similar style lie closely in that space. Since the definition of image style in neural style transfer literature is general and is not dependent on a specific image domain, encoding an image $I$ to its style embedding can also be seen as a general task that is independent of the output domain $B$. This allows for performing the pretraining stage only once using auxiliary training data. The finetuning stage eventually tweaks the embedding to better suit the specific target domain $B$. We show experimentally in Section 4 that pretraining the style encoder on datasets other than the target domain $B$ doesn't degrade the performance. It can even improve the performance if the target dataset is small, in which case pretraining on an auxiliary dataset helps with the generalization of the overall model.

## 4 EXPERIMENTAL EVALUATION

**Datasets.** We evaluate our approach on five standard I2I translation benchmarks used in Isola et al. (2017); Zhu et al. (2017b); Architectural labels → photo, aerial → map, edges → shoes/handbags and night → day. In addition, we use the Space Needle timelapse dataset (Martin-Brualla, 2007), which consists of 2068 paired images with a $8280 \times 1080$ resolution, where the input domain includes images with temporally smoothed appearance, and the output domain contains real images spanning different lighting and weather conditions.

**Baselines.** While we report numbers for retrained models using the code released with BicycleGAN (BicycleGAN v0) for completeness, we mainly compare to two stronger baselines:

- **BicycleGAN v1**: we implement BicycleGAN using the same network architecture as used in our approach to have a fair comparison.

- **BicycleGAN v2**: we augment BicycleGAN with the cross-cycle consistency constraint introduced in (Huang et al., 2018; Lee et al., 2018) as follows: the input is a pair of training examples $(I_1^A, I_1^B), (I_2^A, I_2^B)$ for which we obtain their respective style embeddings $z_1 = E(I_1^B), z_2 = E(I_2^B)$. We then apply a 2-step cyclic reconstruction of $I_1^B, I_2^B$; in the first step, we generate both images with a swapped style $u = G(I_1^A, z_2), v = G(I_2^A, z_1)$. In the the second step, we re-capture the latent style vectors $\hat{z}_2 = E(u), \hat{z}_1 = E(v)$ and generate the original images $I_1^B, I_2^B$ by swapping the style again: $\hat{I}_1^B = G(I_1^A, \hat{z}_1)$, $\hat{I}_2^B = G(I_2^A, \hat{z}_2)$. We add a cyclic reconstruction term for $\hat{I}_1^B, \hat{I}_2^B$.

### 4.1 EVALUATION

**Image reconstruction.** We report the reconstruction quality of validation set images, using both *PSNR* and *AlexNet*-based *LPIPS* (Zhang et al., 2018) metrics, in Table 1. Note that our results without fine-tuning (stage 2) are on-par-with the baselines, which verifies the validity of our approach and

Table 1: Validation set reconstruction quality, as measured by *PSNR* (higher is better) and *LPIPS* (Zhang et al., 2018) (lower is better), for various datasets. We compare between retraining BicycleGAN (Zhu et al., 2017b) authors' released code (Bicycle v0), our implementation of the two baselines (BicycleGAN v1 and BicycleGAN v2) described in Section 4, and our approach both before finetuning (ours - stage 2), and after finetuning (ours - stage 3).

| | edge2shandbags | | edges2shoes | | labels2facades | | night2day | | maps | | space needle | |
|---|---|---|---|---|---|---|---|---|---|---|---|---|
| | PSNR ↑ | LPIPS ↓ | PSNR ↑ | LPIPS ↓ | PSNR ↑ | LPIPS ↓ | PSNR ↑ | LPIPS ↓ | PSNR ↑ | LPIPS ↓ | PSNR ↑ | LPIPS ↓ |
| Bicycle v0 | 17.08 | 0.255 | 20.24 | 0.177 | 12.64 | 0.431 | 13.25 | 0.520 | 14.32 | 0.396 | – | – |
| Bicycle v1 | 18.52 | 0.198 | 21.84 | 0.124 | 13.08 | 0.378 | 13.88 | 0.491 | 14.67 | 0.359 | 19.72 | 0.233 |
| Bicycle v2 | **19.23** | 0.192 | 22.51 | 0.132 | 13.36 | 0.375 | 14.48 | 0.480 | **16.17** | 0.407 | 19.84 | 0.238 |
| ours - stage 2 | 18.01 | 0.209 | 21.40 | 0.140 | **13.44** | 0.383 | 14.34 | **0.476** | 15.08 | 0.392 | **21.39** | 0.227 |
| ours - stage 3 | 18.91 | **0.177** | 22.68 | **0.117** | **13.44** | **0.370** | 15.05 | 0.452 | 15.15 | **0.349** | 22.11 | **0.187** |

Table 2: Generalization of a pretrained style encoder $E$. We report validation set reconstruction for the edges2handbags and night2day datasets when pretraining with different datasets. Stages 2, 3 show results before/after finetuning $E$ respectively.

| Dataset | pretrain dataset | Stage 2 | | Stage 3 | |
|---|---|---|---|---|---|
| | | PSNR ↑ | LPIPS ↓ | PSNR ↑ | LPIPS ↓ |
| edges2handbags | edges2handbags | 18.01 | 0.209 | 18.91 | 0.177 |
| | edges2shoes | 17.89 | 0.215 | 18.96 | 0.176 |
| | space_needle | 17.86 | 0.221 | 19.02 | 0.175 |
| night2day | night2day | 13.75 | 0.489 | 15.15 | 0.454 |
| | space_needle | 14.34 | 0.476 | 15.05 | 0.452 |
| | edges2handbags | 13.91 | 0.492 | 15.03 | 0.461 |

Table 3: Diversity score is the average LPIPS distance (Zhang et al., 2018). User preference score is the percentage a method is preferred over Ours v4, on the edges2shoes dataset.

| Approach | LPIPS ↑ (transfer) | LPIPS ↑ (sampling) | User preference ↑ |
|---|---|---|---|
| Bicycle v1 | 0.102 | 0.119 | 30.0% |
| Bicycle v2 | 0.138 | 0.132 | 37.7% |
| Ours v1 | 0.153 | 0.148 | 46.5% |
| Ours v2 | 0.171 | 0.140 | 41.1% |
| Ours v3 | 0.149 | 0.165 | 50.4% |
| Ours v4 | 0.154 | 0.132 | 50% |

that style-based encoder pre-training successfully learns to distinguish different modes in the output domain, which proves effective for training multi-modal I2I networks. Fine-tuning (stage 3) further improves our results compared to the baselines. Figure 2 shows qualitatively how our approach reconstructs the target style more faithfully.

**Style transfer and sampling.** Figure 3 shows style transfer to validation set images from different datasets. We can also sample random styles directly from the the latent distribution as described in Section 3.2. Figure 5 shows results for both adhoc sampling from the assumed $N(\mu, \sigma)$ empirical distribution, as well as formally sampling from a unit Gaussian using the mapper network $\mathcal{M}$. While both results look good, we note that the assumption for adhoc sampling is not explicitly enforced, and thus could sample bad style codes outside the distribution (see Appendix A.5 for examples).

**Style interpolation.** Figure 4 shows style interpolation by linearly interpolating between two latent vectors. Note the smooth change in lighting and cloud patterns when going from cloudy to sunny (second row).

**Pre-training generalization.** Since the notion of style, as defined in the Neural Style Transfer literature, is universal and not specific to a certain domain, we hypothesized that style-based encoder pretraining would learn a generic style embedding that can generalize across multiple domains and be effective for multi-modal image I2I translation. Here, we experimentally verify our hypothesis in Table 2. For a target dataset, we train the generator $G$ three times, each with different pre-training of the style encoder $E$: (1) same dataset pre-training: pre-train $E$ using the output domain $B$ of the target dataset. (2) similar-domain pre-training: pre-train on a different dataset, but whose output domain bears resemblance to the output domain of the target dataset (e.g., edges2shoes and edges2handbags, or day images from night2day and the Space Needle timelapse dataset). (3) different-domain pre-training: pre-train on a different dataset whose output domain has different styles from that of the target dataset (e.g., edges2handbags and the Space Needle timelapse datasets, or night2day and edges2handbags datasets). Table 2 shows that without fine-tuning (stage 2), the edges2handbags dataset shows a slight performance degradation when going from pre-training on the same dataset, to pre-training on a similar-domain dataset, and finally pre-training on a different-domain dataset. On the other hand, the night2day dataset has only ∼100 unique scenes for training. So, pre-training on another dataset such as Space Needle generalizes better to new scenes in the validation set, since it helps avoid overfitting the small number of unique scenes in the training set. After fine-tuning,

Table 4: Ablation study on the effect of different components and loss terms using the edges2handbags dataset. We study direct and cyclic reconstructions on ground truth images (dir_recon, cyc_recon), discriminator loss on direct reconstructions and on generated images with a randomly sampled style (D_dir, D_rand_z), latent reconstruction (z_recon), $L2$ and KL regularization on the latent vector $z$ (z_L2, z_KL), and finally the use of VAE vs. just an auto-encoder.

| Approach | Loss setup | | | | | | | | PSNR↑ | IS↑ | LPIPS ↓ |
|---|---|---|---|---|---|---|---|---|---|---|---|
| | dir_recon | cyc_recon | D_dir | D_rand_z | z_recon | z_L2 | z_KL | VAE | | | |
| Bicycle v1 | ✓ | – | ✓ | ✓ | ✓ | – | ✓ | ✓ | $18.28 \pm 0.30$ | $2.31 \pm 0.05$ | $0.201 \pm 0.003$ |
| Bicycle v2 | ✓ | ✓ | ✓ | ✓ | ✓ | – | ✓ | ✓ | $18.96 \pm 0.30$ | $2.45 \pm 0.07$ | $0.192 \pm 0.002$ |
| Bicycle v1.2 | ✓ | – | ✓ | ✓ | ✓ | – | ✓ | – | $19.02 \pm 0.10$ | $2.36 \pm 0.12$ | $0.175 \pm 0.001$ |
| Bicycle v2.2 | ✓ | ✓ | ✓ | ✓ | ✓ | – | ✓ | – | $19.34 \pm 0.07$ | $2.44 \pm 0.06$ | $0.176 \pm 0.002$ |
| Bicycle v1.3 | ✓ | – | ✓ | ✓ | ✓ | ✓ | – | – | $19.21 \pm 0.06$ | $2.34 \pm 0.08$ | $0.177 \pm 0.002$ |
| Bicycle v2.3 | ✓ | ✓ | ✓ | ✓ | ✓ | ✓ | – | – | $19.24 \pm 0.09$ | $2.33 \pm 0.04$ | $0.180 \pm 0.001$ |
| Ours v1 | ✓ | ✓ | ✓ | ✓ | ✓ | ✓ | – | – | $18.97 \pm 0.13$ | $2.41 \pm 0.07$ | $0.189 \pm 0.004$ |
| Ours v2 | ✓ | – | ✓ | ✓ | ✓ | ✓ | – | – | $18.94 \pm 0.10$ | $2.43 \pm 0.03$ | $0.183 \pm 0.002$ |
| Ours v3 | ✓ | – | ✓ | – | – | ✓ | – | – | $18.94 \pm 0.05$ | $2.42 \pm 0.03$ | $0.176 \pm 0.001$ |
| Ours v4 | ✓ | – | ✓ | – | – | – | – | – | $18.94 \pm 0.02$ | $2.46 \pm 0.03$ | $0.177 \pm 0.001$ |

performance differences further reduce to be insignificant. We also investigate the generalization of encoder pre-training using non-style distance metrics in Appendix A.4.

**Ablative study.** We investigate the role of different loss terms as we transition from the loss setup of baselines to that of our training approach. We first remove the variational part in both BicycleGAN v1 and BicycleGAN v2 baselines resulting in Bicycle v1.2, v2.2. We further remove the Gaussian prior and replace the KL loss with an L2 regularization in Bicycle v1.3, v2.3. To maintain random latent vector sampling during training without a prior, we sample a random training image, and use its style code. We define different versions of our approach (v1, v2, v3, and v4) based on different loss setup during training as follows: we start with 'Ours v1', which has the same setup as Bicycle v2.3, except that it uses pre-trained embeddings as described in Section 3.1. We then remove cyclic reconstruction, random z sampling, and L2 regularization terms resulting in 'Ours v2', 'v3', and 'v4' respectively. We run each setup on the edges2handbags dataset. In order to draw more reliable conclusions, we repeat each experiment 3 times and report the mean and standard deviation in Table 4. We notice that removing the variational part in VAEs is enough to achieve good results. While VAEs in general are robust to noise in the input latent, we observe that this comes at the expense of the expressiveness of the latent space (e.g., less faithful style capture and transfer), especially for low dimensional latents. We also observe that our approach generally performs better with less constraints (loss terms). For example, "Ours v1, v2" have lower results than their "Bicycle v1.3, v2.3" counterparts. This shows that the main benefit of pre-trained embeddings is when the network is less constrained.

**Diversity and user study.** We evaluate diversity by computing the average LPIPS distance over 1600 output images. We measure diversity on two setups: we sample 100 validation images, and (1) apply style transfer from 16 randomly sampled images, or (2) we sample 16 random codes using the mapper network $\mathcal{M}$ to obtain 1600 outputs. We also measure the realism and faithfulness of style transfer through a user study, where 30 participants are shown an input shoe sketch, an input style image and two style transfer outputs. They are given unlimited time to choose which output looks more realistic, and if both are realistic, then which transfers the style more faithfully. We fix 'Ours v4' approach as anchor and compare other methods to it. Table 3 shows that the baselines achieve lower diversity and user preference compared to our approach, specially in the style transfer setup. Different variations of our method, except for 'Ours v2' yield similar diversity and user preference scores. We observe that 'Ours v2' shows artifacts in some outputs, leading to higher diversity but lower user preference. Our diversity results for the style sampling setup have some variation and are sensitive the mapper network training, but are still either on-par or better than the baselines.

**Conclusion.** We investigated the effectiveness of embedding pre-training for the task of multi-modal I2I translation. The pre-training can be done once on auxiliary data, and generalizes well to other domains. This allows for a faster training of I2I translation networks with fewer losses and achieves more faithful style capture and transfer. Furthermore, we studied the contribution of different loss terms, where we discovered that noise added by a variational auto-encoder can limit the expressiveness of low-dimensional latent spaces. Finally, we achieved state-of-the-art results on several benchmarks.

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

# A   APPENDIX

## A.1   IMPLEMENTATION DETAILS

The generator network $G$ has a symmetric encoder-decoder architecture based on Wang et al. (2018a), with extra skip connections by concatenating feature maps of the encoder and decoder. We use a multiscale-patchGAN discriminator (Wang et al., 2018a) with 3 scales and employ a LSGAN (Mao et al., 2017) loss. The mapper network $\mathcal{M}$ is a multi-layer perceptron (MLP) with three 128-dimensional hidden layers and a *tanh* activation function. For the reconstruction loss, we use the perceptual loss (Johnson et al., 2016) evaluated at $\texttt{conv}_{i,2}$ for $i \in [1, 5]$ of VGG (Simonyan & Zisserman, 2014) with linear weights of $w_i = 1/2^{6-i}$ for $i \in [1, 5]$. The architecture of the style encoder $E$ is adopted from Lee et al. (2018), and we use a latent style vector $z \in \mathbb{R}^8$. Our optimizers setup is similar to that in Zhu et al. (2017b). We use three *Adam* optimizers: one for the generator $G$ and encoder $E$, another for the discriminator $D$, and another optimizer for the generator $G$ alone with $\beta_1 = 0, \beta_2 = 0.99$ for the three optimizers, and learning rates of $0.001, 0.001$ and $0.0001$ respectively. We use a separate *Adam* optimizer for the mapper network $\mathcal{M}$ with $\beta_1 = 0.5, \beta_2 = 0.99$, and a learning rate of $0.01$ with a decay rate of $0.7$ applied every 50 steps. Relative weights for the loss terms are $\lambda_{\text{cGAN}} = 1$, $\lambda_{\text{rec}} = 0.02$ and $\lambda_{L2} = 0.01$ for the GAN loss, reconstruction loss, and $L2$ latent vector regularization respectively. When sampling triplets for any anchor image $I_c$, we use $k_c = 5, k_f = 13$ for the size of the set of close and far neighbors respectively.

## A.2   MORE QUANTITATIVE COMPARISON

We report the Inception Score (IS) computed over the validation set of various datasets in Table 5. Surprisingly, results after finetuning ("ours - stage 3") are slightly worse than those before finetuning ("ours - stage 2"), but both are still better than the baselines except for the maps dataset. We also note the Inception Score is not very suited to image-to-image translation tasks, since it prefers output diversity with respect to ImageNet classes, not within-class diversity as in our case.

Table 5: Inception score comparison (higher is better) for different datasets.

|  | handbags | shoes | facades | night2day | maps | space needle |
|---|---|---|---|---|---|---|
| Bicycle v1 | 2.13 | 2.83 | 1.41 | 1.65 | 3.26 | 1.82 |
| Bicycle v2 | 2.07 | 2.64 | 1.45 | 1.74 | **3.57** | 1.77 |
| ours - stage 2 | **2.22** | **2.75** | **1.61** | 1.76 | 3.32 | **1.90** |
| ours - stage 3 | 2.15 | 2.85 | 1.56 | **1.84** | 3.28 | 1.89 |

## A.3   LATENT SPACE VISUALIZATION

Figure 6a visualizes the latent space learned by the style encoder $E$ after pretraining and before finetuning (a), after finetuning (b), and the latent space learned by BicycleGAN (Zhu et al., 2017b) (c). The embedding learned through pretraining and before finetuning shows meaningful clusters. Finetuning further brings the embedding closer to that of BicycleGAN.

## A.4   ENCODER PRE-TRAINING WITH NON-STYLE METRICS

Pre-training the encoder using a style-based triplet loss showed to be successful for multi-modal image translation tasks where the variability in the target domain is mainly color-based. This is shown in the results obtained on several benchmarks, even before the finetuning stage ("ours - stage 2" in Table 1). We note though that the usage of style-loss as a distance metric for triplet sampling is just one choice and can be replaced with other distance metrics depending on the target application. Triplet sampling with style distance results in learning an embedding space where images with similar colors/styles lie closely in that space as shown in Section A.3. If, for example, we sample triplets instead based on the distance between VGG-Face (Parkhi et al., 2015) embeddings, the encoder will learn a latent space which is clustered by identity. In this section, we aim to validate that the proposed pre-training strategy can be extended to multi-modal image-to-image translation tasks with non-style variability. We inspect the task of manipulating facial expressions, where the input is a

neutral face, and the output can have other emotions or facial expressions. For this task, similar emotions should be embedded closely in the latent space. We therefore use an off-the-shelf facial expression recognition system to compute the emotion similarity/distance between any pair of images. Specifically, we compute the emotion distance as the euclidean distance between the 512-dimensional feature map of the last layer of a pretrained classification network (e.g., (Jie, 2018)). We visualize the learned latent space in Figure 7, which shows clusters with similar emotions or facial expressions. We also show example translation results on a holdout set of the front-view images of the KDEF dataset (KDEF, 2017) in Figure 8. We note that the generator successfully learns to manipulate facial expressions based solely on the pre-trained embeddings (without the finetuning stage). On the other hand, the BicycleGAN-based baselines collapsed to a single mode (over 3 different runs). This shows that our staged-training approach is stable and not sensitive to hyper-parameters, unlike the BicycleGAN baselines which will require careful hyper-parameter tuning to work properly on this task. We also point out that the poor output quality is mainly due to using a pixel-wise reconstruction loss for the generator training, while the input-output pairs in this dataset are not aligned. We didn't investigate improving the generator training as this is orthogonal to verifying the generalization of encoder pre-training.

## A.5 STYLE SAMPLING COMPARISON

Figure 9 compares style sampling using the mapper network $\mathcal{M}$ vs adhoc sampling from the assumed $N(\mu, \sigma)$ of an $L2$-regularized latent space, where $\mu, \sigma$ are empirically computed from the training set. Note that adhoc sampling can sometimes sample bad style codes outside the distribution (e.g. third image in first row, and first image in third row in the right side of Figure 9), since the assumption that a $L2$-regularized space would yield normally distributed latents with zero mean and low standard deviation is not explicitly enforced.

## A.6 TRAINING TIME

Simplifying the training objective allows for faster training time, as well as a larger batch size due to lower memory usage. Table 6 shows the processing time per 1000 training images for the baselines as well as different variations of our approach as defined in Table 4.

Table 6: Training time (in seconds) per 1000 images for the baselines, as well as different versions of our approach (defined in Table 4).

| Approach | Batch size | time/kimg↓ (sec) | Max batch size | time/kimg↓ (sec) |
|---|---|---|---|---|
| Bicycle v1 | 8 | 93.11 | 12 | 85.36 |
| Bicycle v2 | 8 | 155.72 | 8 | 155.72 |
| Ours v1 | 8 | 145.50 | 8 | 145.50 |
| Ours v2 | 8 | 98.55 | 12 | 93.04 |
| Ours v3 | 8 | 64.92 | 16 | 53.80 |
| Ours v4 | 8 | 63.96 | 16 | 51.23 |

## A.7 CONVERGENCE ANALYSIS

Figure 10 compares the convergence of our staged training compared to the BicycleGAN baselines. The dotted line in the graph marks the transition between stages 2 and 3 of our training (i.e, switching from a fixed pre-trained encoder $E$ to finetuning both $G$ and $E$ together). We measure the reconstruction error (LPIPS) of the validation set of the edges2handbags dataset as the training progresses. Results show that with a fixed pre-trained encoder, staged training starts with higher error than the baselines, but quickly drops to show similar performance as the baselines, and even beats the baselines before switching to stage 3 (marked by a dotted line). When starting to finetune the encoder $E$, we get a spike in the reconstruction error as the network adapts to the shift in the pre-trained embeddings, but then the performance of the staged training steadily widens the performance gap with the baselines.

This shows the importance of the finetuning stage to tweak the pre-trained embeddings to better serve the image synthesis task.

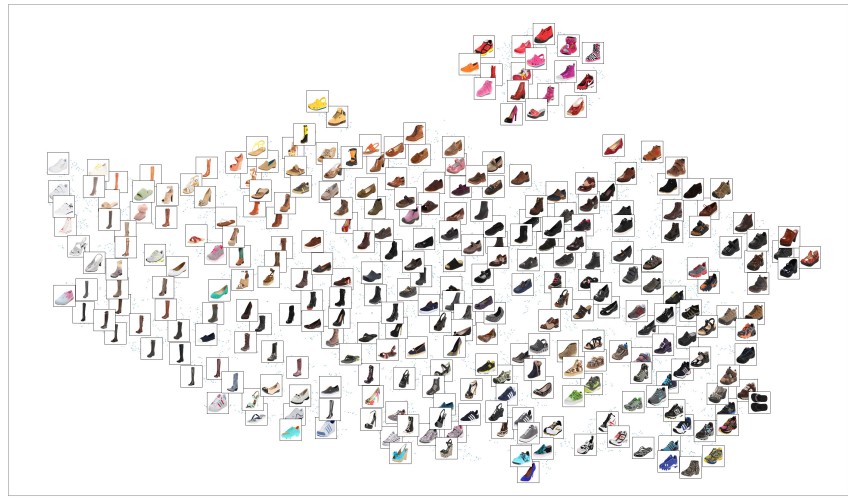

(a) Our approach: after style pretraining.

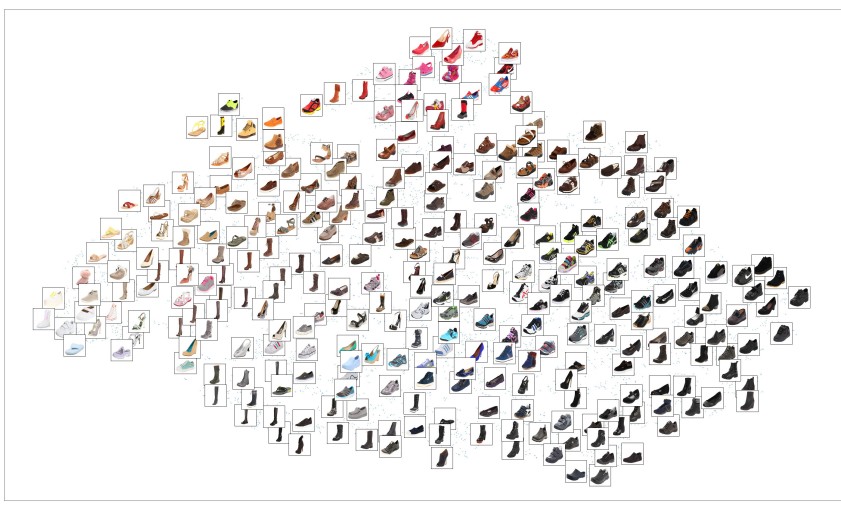

(b) Our approach: after finetuning.

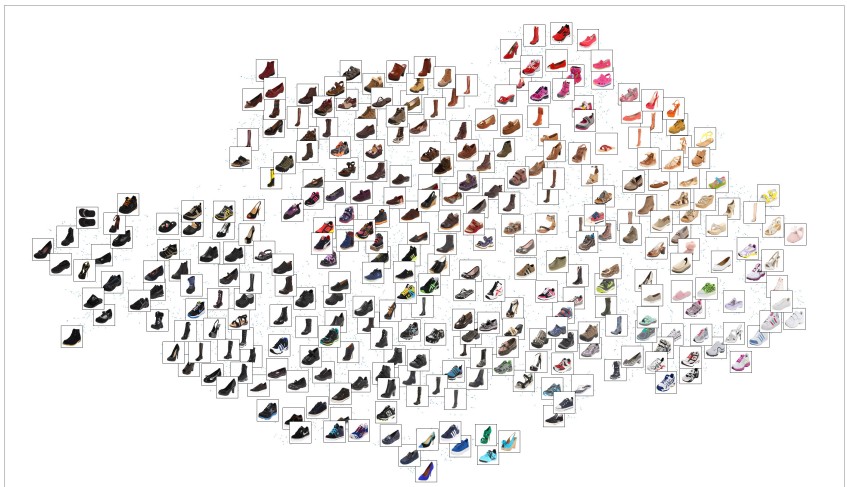

(c) BicycleGAN v1 baseline.

Figure 6: t-SNE plots for the latent style space learned by the style encoder $E$ (a) after style pretraining, (b) after finetuning, and (c) using the BicycleGAN v1 baseline.

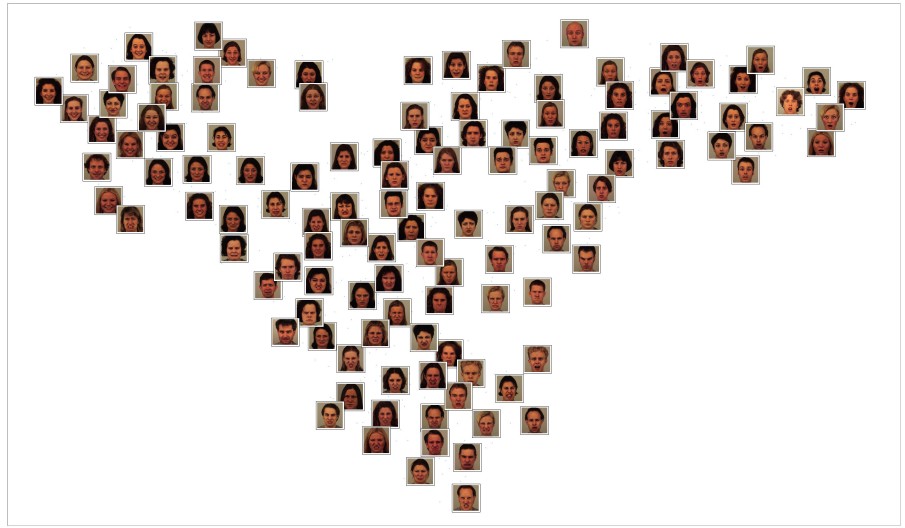

Figure 7: t-SNE plot for the pre-trained latent space learned for facial expressions on a subset of the KDEF dataset.



Figure 8: Emotion translation results. First row shows the input image, as well as the ground truth images from which we encode the latent emotion vector for reconstruction. Our staged training approach is able to achieve multi-modal synthesis, while the baselines collapse to a single mode.

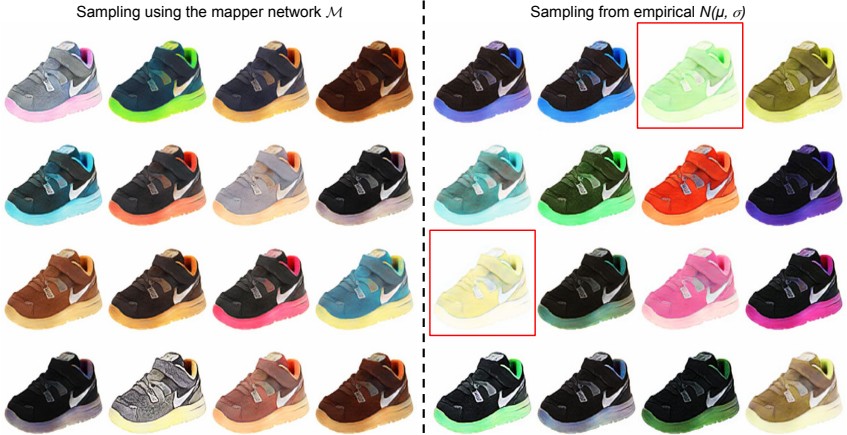

Figure 9: Sixteen randomly sampled styles using both the mapper network $\mathcal{M}$ (left), as well as adhoc sampling from the empirically computed $N(\mu, \sigma)$ distribution of a $L2$-regularized latent space (right). Adhoc sampling could sample bad style codes outside the latent distribution (marked in red).

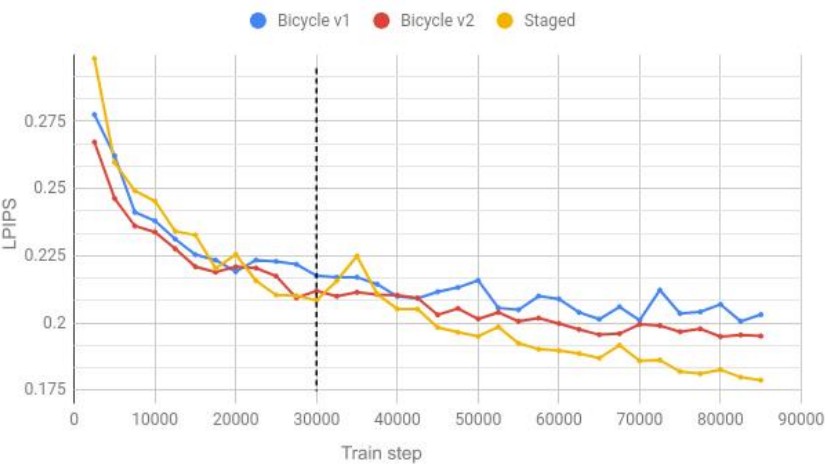

Figure 10: Convergence comparison between the proposed staged training (ours - v3) and the BicycleGAN baselines measured by the reconstruction error (LPIPS) of the validation set of the edges2handbags dataset. Dotted line shows the transition between stages 2 and 3 of our training (i.e, switching from a fixed $E$ to finetuning both $G$ and $E$ together).

