# OpenReview forum: "Style-based Encoder Pre-training for Multi-modal Image Synthesis"
_ICLR.cc/2020/Conference — Reject_

### Official Review · AnonReviewer1 · 2019-10-23
**Official Blind Review #1**

**Rating:** 3

**Review:**

This paper tackles the Image-to-Image translation task via a simplified yet more effective training procedure.

Compared to the direct baseline BicycleGAN, the training procedure proposed in this paper replaces the simultaneous training of the
encoder E and the generator G with a staged training that alternatively trains on E and G and then finetune them together. Although this appears to be a simple modification, the empirical performance for generalization and reconstruction qualities prove the effectiveness of the proposal.

It is better to provide more intuition on why this pretraining phase would help to make the results generalize better and yield better performance. The current presentation of the paper mostly consists of detailed descriptions of the proposal training procedure, without some interesting discussions about why this pretraining makes the problem easier. For instance, I'm interested in seeing with some toy distributions, what is the training progress (measured quantitatively) comparing the proposed method and traditional BicycleGAN.

Although the results look nice, with the current presentation, there's not much inspiration one could get from the paper. I encourage the authors to make some adjustments, and I will reconsider the score.

**Experience Assessment:**

I have read many papers in this area.

**Review Assessment: Checking Correctness Of Derivations And Theory:**

I assessed the sensibility of the derivations and theory.

**Review Assessment: Checking Correctness Of Experiments:**

I assessed the sensibility of the experiments.

**Review Assessment: Thoroughness In Paper Reading:**

I read the paper at least twice and used my best judgement in assessing the paper.

---

> ### Author Response · Authors · 2019-11-14
> **Response to blind review #1**
>
> Thank you for reviewing our paper! We address your concerns one by one below:
>
> Intuition why the pretraining phase makes the training easier:
>   - Encoder pretraining with a triplet loss learns an embedding space that models the distribution modes into clusters defined by the distance metric used for computing the triplet loss (Figure 6-a shows clusters of similar styles and Figure 7 shows clusters of facial expressions).
>   - During stage 2, the latent space is kept fixed, and the input latent to the generator can be used to clearly distinguish the style cluster to which the output belongs, which makes the multi-modal synthesis task easier for the generator. It also helps avoid mode collapse when the variability in the output domain is not very obvious to the encoder during simultaneous training of G and E. We observed that BicycleGAN collapses to a single mode when training for the task of emotion transfer (section A.4), and also during some runs on the Maps dataset (retraining BicycleGAN with a lower learning rate eventually worked for the Maps dataset). On the other hand, our approach is stable and we didn’t need to change any hyperparameters between different datasets, except for using a smaller network (number of filters) for small datasets like Labels2Facades and Night2day to avoid overfitting).
>   - Results before finetuning (Table 1: “ours - stage 2” row) quantitatively show that pre-training alone (without finetuning) leads to reasonable results on-par with BicycleGAN, which proves the validity of our approach.
>   - Stage 3 which finetunes both G and E together allows for tweaking the learned embedding to better suit the synthesis task, which reflects on further performance improvement in Table 1. Figure 6-a and 6-b visualize the latent space before and after finetuning. The finetuned latent looks smoother and we observed that the covariance matrix after finetuning becomes more like a Gaussian.
>
> Intuition why pretraining generalizes across datasets:
> - Why Gram matrices? The usage of Gram matrices for neural style transfer proved to be very effective and it reliably captures the style of arbitrary input style images. This implies that Gram matrices can reliably encode the styles from a wide range of domains, and they are not very specific to certain domains.
> - This led to our hypothesis that encoder pretraining using a style-based triplet loss would learn a generic style embedding that can generalize across multiple domains and be effective for multi-modal image I2I translation. We experimentally validated this hypothesis in Table 2 of the paper.
> - We however note that while style-based pretraining works well for many different domains where the output variability is mainly color and texture based (for example the 6 datasets in Table 1), it is not suitable when the domain variability is not color-based (e.g., facial expressions). But, we also show preliminary results for encoder pre-training using non-style metrics in section A.4, which is able to achieve multi-modal image synthesis while the Bicycle-GAN baseliens collapse to a single mode.
>
> Why is this work useful to the community:
> - We believe that while end-to-end training poses a convenience, sometimes it can be advantageous both in terms of training stability and output quality for downstream tasks to break down the training into proxy tasks. We believe our work sheds some light on the usefulness of pre-training for image synthesis tasks, which aligns with the standard training paradigm in visual recognition of first pretraining on a proxy task, either large supervised datasets (e.g., ImageNet) or unsupervised tasks and then fine-tuning (transfer learning) on the desired task.
>
> The current presentation of the paper lacks discussion: we agree that we detailed our approach and shortened the discussion due to space limit. We propose the following changes to make space for the discussion points mentioned in our response above (we will submit a revision of the paper before the end-of-rebuttal period).
> - Move Table 4 about training speed and its corresponding paragraph to the supplementary materials
> - Move the details of the user study (the paragraph before “conclusion” to the supplementary materials.
> - Make Figures smaller (e.g., figures 1 and 5).
>
> Quantitative evaluation for how the training progresses compared to BicycleGAN: with toy or small datasets (e.g. Labels2Facades), the problem becomes easy and the benefits of the proposed pre-training become less clear. We instead propose to include convergence analysis on one of the larger datasets (e.g., edges2handbags, edges2shoes or space needle) where we observe that our approach learns to correctly match the style of output images both faster and more accurate than BicycleGAN. Specifically, we will report quantitative metrics (e.g, LPIPS, L1, PSNR) over the validation set as the training progresses.

---

### Official Review · AnonReviewer3 · 2019-10-23
**Official Blind Review #3**

**Rating:** 3

**Review:**

In this paper, the authors tackle the problem of multi-modal image-to-image translation by pre-training a style-based encoder. The style-based encoder is trained with a triplet loss that encourages similarity between images with similar styles and dissimilarity between images with different styles. The output of the encoder is a style embedding that helps differentiates different modes of image synthesis.  When training the generator for image synthesis, the input combines an image in the source and a style embedding, and the loss is essentially the sum of image conditional GAN loss and perceptual loss. Additionally, the authors propose a mapping function to sample styles from a unit Gaussian distribution.

I think the idea of pre-training a style-based encoder is straightforward. I am mainly concerned about the performance of the presented approach. First, there are no many visual comparisons in the paper. The only visual comparison is in Figure 8, but results are only limited to faces. The visual results in Figure 5 do not look appealing to me. The change in the style mainly comes from the global change in color: no much change in the texture or local color. The "night2day" results look poor to me.  I am concerned about the diversity of the styles learned in the model.

On the other hand, I am convinced that the proposed model is better than BicycleGAN, and the approach is somehow novel. The user study in Table 5 suggests that the proposed method is somehow better than BicyleGAN in visual quality on one task. My overall rating is borderline.

Minor comments:
- In the first sentence of Section 3.2, I do not think "one-to-one correspondence" is the right description. The encoder is not expected to be invertible.
- In Equation (3), "e_i" is a little bit misleading. It does not mean the i-th element in {"e_j"}. You may want to replace "e_i" with "s_i" to avoid confusion.
- The explanation of ours v1, v2, v3, v4 is not clear. It is also difficult to find its definition.

**Experience Assessment:**

I have published one or two papers in this area.

**Review Assessment: Checking Correctness Of Derivations And Theory:**

I assessed the sensibility of the derivations and theory.

**Review Assessment: Checking Correctness Of Experiments:**

I assessed the sensibility of the experiments.

**Review Assessment: Thoroughness In Paper Reading:**

I read the paper at least twice and used my best judgement in assessing the paper.

---

> ### Author Response · Authors · 2019-11-14
> **Response to blind review #3**
>
> Thank you for reviewing our paper! Please find responses to your points/questions below:
>
> Visual comparisons: Figure 2 in the paper shows qualitative comparisons, not just figure 8. Also, Table 5 shows results of a user study which compared the quality of 50 different outputs of our method against the BicycleGAN-based baselines.
>
> Diversity comes from global change in color: we would like to highlight that there is plenty of evidence in the paper that the diversity doesn’t come from just the global color change. For example:
> - Figure 5 (style sampling):
>     - Space needle: results show clear weather changes, such as cloudy vs sunny, change in cloud patterns and even sampling foggy weather which was present in some images in the training set.
>     - Maps dataset: the existence and/or density of bushes clearly varies between different sampled styles.
>     - Edges2handbags: the texture of the bag varies clearly between smooth and rough leather (better seen in zoom).
> - Figure 4 (style interpolation):
>     - The smooth change from cloudy to sunny weather including the change in lighting and cloud patterns.
> - Figure 3 (style transfer):
>     - Space needle: the output style clearly copies the weather condition. We show sunset, sunny, foggy and cloudy weather.
>     - Night2day: variation in lighting conditions can be clearly seen including transferring whether the surface is sunlit or not, as well as different cloud patterns and clear skies.
>     - Edges2handbags: the output texture varies (most clear in the third image with the dark pink color). Figure best seen in zoom.
> - Figure 2 (qualitative comparison):
>     - Our approach matches the GT texture better than the baselines (most obvious in the left column - first and third rows) — figure best seen in zoom.
> Also, we quantitatively measure the output diversity in both style transfer and style sampling setups in Table 5.
>
> Quality of results:
> We recommend looking at the figures in zoom to better appreciate the quality. The night2day and labels2facades datasets are particularly smaller than the other datasets, as night2day has only 100 unique scenes in the training set, while the labels2facades trainset contains only a few hundred images. We observed that our models overfits the training set if we trained with full capacity like the larger datasets. While we used the same hyper-parameters for all datasets, the change we made for training on the night2day and labels2facades datasets is to use a smaller model (i.e reduce the number of convolution filters), and train for shorter (i.e early stopping).
>
> Minor comments: thank you for the suggestions. We will adopt the suggested changes, and include a clear definition for the different versions of our method, and we will submit a revised version of our paper by Friday.

---

### Official Review · AnonReviewer2 · 2019-11-04
**Official Blind Review #2**

**Rating:** 3

**Review:**

Summary:
The authors propose to use a non-end-to-end approach to the problem of multi-modal I2I. Firstly, a metric learning problem is solved to embed images into space, taking into account the pairwise style discrepancy (style is defined, e.g., based on VGG Gramians). As the notion of style is universal for similar datasets, this step further is shown to be generalizable. Secondly, the generator is trained on a supervised image translation tasks: the original image and the style, extracted from the target image, are fed to the generator, and the output is a translated image. Thirdly, style encoder and generator are simultaneously finetuned.

Overall, this is an incremental work in the field of supervised I2I.

Questions:
1. Is it true that the proposed approach requires semantically aligned datasets, as the style of the whole image is described with a comparatively low-dimensional vector, and the GAN objective is applied to paired outputs only? Compare, e.g., with Gramian-based style transfer, where segmentation masks are often desired for better results [1].
2. Can the developed pipeline be generalized to the unsupervised setting, e.g., involving a cycle consistency loss and a non-conditional GAN objective? To my mind, such generalization can show the greater importance of the described method.

Remarks:
1. Formula (1) is incorrect. I guess the first term should contain z instead of g. Otherwise, the encoder parameters are optimized using the regularizer only.

[1] Jaejun Yoo, Youngjung Uh, Sanghyuk Chun, Byeongkyu Kang, Jung-Woo Ha. Photorealistic Style Transfer via Wavelet Transforms. ICCV 2019.

**Experience Assessment:**

I have read many papers in this area.

**Review Assessment: Checking Correctness Of Derivations And Theory:**

I carefully checked the derivations and theory.

**Review Assessment: Checking Correctness Of Experiments:**

I assessed the sensibility of the experiments.

**Review Assessment: Thoroughness In Paper Reading:**

I read the paper at least twice and used my best judgement in assessing the paper.

---

> ### Author Response · Authors · 2019-11-14
> **Response to blind review #2**
>
> Thank you for reviewing our paper! Please find responses to your points/questions below:
>
> We believe our work is orthogonal to existing works in the I2I translation filed. We focus on the importance of pre-training for multi-modal image synthesis, which has not been explored in the I2I translation field before. Our work provides insights that embedding pre-training improves the training stability and the overall output quality and diversity, as well as simplifying the generator training by using fewer losses. In addition we show that pre-training generalizes well across datasets, which is specifically useful to the case of having limited training data.
>
> Specifically we show both quantitatively and qualitatively that pre-training provides the following advantages:
> - More stable training: pre-training helps avoid mode collapse when the variability in the output domain is not very obvious to the encoder (E) during simultaneous training of G and E. We observed that BicycleGAN collapses to a single mode when training for the task of emotion transfer (section A.4), and also during some runs on the Maps dataset (retraining with a lower learning rate eventually worked for the Maps dataset). On the other hand, our approach is stable and we didn’t need to change any hyperparameters between different datasets, except for using smaller number of filters for small datasets like Labels2Facades and Night2day to avoid overfitting).
> - Generalization of the pre-training stage (Table 2) is an important property and improves the results when training data is limited (as observed in the night2day dataset which only contains ~100 unique scenes in the training set).
> - Better output quality (quantitative results in Table 1, qualitative comparison in Figure 2 and user study in Table 5).
> Simplified and faster training of the generator: Table 3 shows that we can maintain comparable performance with training with fewer losses than other approaches (e.g, “ours v4” only uses reconstruction and GAN losses). This also reflects on the training speed as shown in Table 4.
> We also provide a thorough study of the importance of different loss terms in Table 3, where we interestingly observed that noise introduced by VAE training can limit the expressiveness of low-dimensional latent spaces.
>
> Is using aligned data a requirement for our approach due to using a low-dimensional latent?
> While we studied pre-training in the supervised setting, we believe that using paired data is not a requirement. We note that some works (e.g., MUNIT [2], DRIT [3]) successfully extended BicycleGAN to the unpaired setting, while maintaining the 8-dimensional latent, through employing a cross-cycle consistency constraint.
> Our work explores the importance of pre-training to differentiate different modes for image synthesis, and hence is orthogonal to supervised vs. unsupervised I2I works like BicycleGAN, MUNIT and DRIT. We are excited to explore the extension to the unsupervised setting like in [2, 3]; however, we believe  that supervised I2I translation is a freestanding problem, and improving the training and results in this setting is of value.
> We however agree with the reviewer that low dimensionality of the style latent in general poses a limit on the amount of information than can be encoded. We chose a dimensionality of 8 to be consistent with previous works (e.g., BicycleGAN), but it is also interesting to study how well our approach can scale to higher latent dimensionality.
>
> Extension to the unsupervised setting:
> We are excited to explore the extension of our approach to the unsupervised setting similar to [2, 3], but we chose to study the effect of pre-training on the supervised setting first (e.g. BicycleGAN setting) before moving to the more challenging unsupervised setting which we left as future work.
>
> Mistake in Formula (1): thank you for catching this, we will fix that in a revised version before Friday!
>
> [2] Xun Huang, Ming-Yu Liu, Serge Belongie, Jan Kautz. Multimodal Unsupervised Image-to-Image Translation.
> [3] Hsin-Ying Lee, Hung-Yu Tseng, Qi Mao, Jia-Bin Huang, Yu-Ding Lu, Maneesh Singh, Ming-Hsuan Yang. DRIT++: Diverse Image-to-Image Translation via Disentangled Representations.

---

### Author Response · Authors · 2019-11-15
**New revision uploaded**

Dear reviewers,

Thank you for your valuable suggestions. We uploaded a new version of the paper with some of the changes we discuss in our individual responses below. In particular, we made the following changes:
(1) included more discussion about:
  - why the proposed staged training is better in terms of training stability and output quality.
  - why style-based pretraining generalizes across datasets.
  - the effectiveness of encoder pre-training for the task of multi-modal image synthesis, which is orthogonal to previous works in the field.
To make space for these changes, we moved Table (4) about the runtime analysis and its discussion to the appendix.
(2) added a "Convergence analysis" section to the appendix with an experiment to quantitatively compare the convergence of the proposed approach compared to the baselines.
(3) fixed a mistake in equation (1)
(4) other proposed suggestions by the reviewers like clarifying the notation in some equations and rephrasing parts of the text, and clarifying some definition.
(5) changed figure 1 to better illustrate our approach.

Thank you

---

### Decision · Program_Chairs · 2019-12-19

**Decision:**

Reject

**Comment:**

The submission describes a new two-stage training scheme for multi-modal image-to-image translation. The new scheme is compared to a single-stage end-to-end baseline, and the advantage of the new scheme is demonstrated empirically. All three reviewers appreciate the proposed contribution and the quality improvement it brings over the baseline. At the same time, the reviewers see the contribution as incremental and not sufficient for an ICLR paper. The author response and paper adjustment have not changed the opinion of the reviewers, so the overall recommendation is to reject.